# Evinacumab, an ANGPTL3 Inhibitor, in the Treatment of Dyslipidemia

**DOI:** 10.3390/jcm12010168

**Published:** 2022-12-25

**Authors:** Bożena Sosnowska, Weronika Adach, Stanisław Surma, Robert S. Rosenson, Maciej Banach

**Affiliations:** 1Department of Preventive Cardiology and Lipidology, Medical University of Lodz, 93-338 Lodz, Poland; 2Faculty of Medical Sciences in Katowice, Medical University of Silesia, 40-752 Katowice, Poland; 3Metabolism and Lipids Unit, Mount Sinai Hospital and Mount Sinai Heart, Icahn School of Medicine at Mount Sinai, New York, NY 10029, USA; 4Cardiovascular Research Centre, University of Zielona Gora, 65-417 Zielona Gora, Poland; 5Department of Cardiology and Adult Congenital Heart Diseases, Polish Mother’s Memorial Hospital Research Institute (PMMHRI), 93-338 Lodz, Poland

**Keywords:** ANGPTL3 inhibitors, evinacumab, familial hypercholesterolemia, LDL cholesterol

## Abstract

Familial hypercholesterolemia (FH) is an inherited disorder. The level of low-density lipoprotein cholesterol (LDL-C) in patients with homozygous FH can be twice as high as that in patients with heterozygous FH. The inhibition of ANGPTL3 shows an important therapeutic approach in reducing LDL-C and triglycerides (TG) levels and, thus, is a potentially effective strategy in the treatment of FH. Evinacumab is a monoclonal antibody inhibiting circulating ANGPTL3, available under the trade name Evkeeza^®^ for the treatment of homozygous FH. It was reported that evinacumab is effective and safe in patients with homozygous and heterozygous FH, as well as resistant hypercholesterolemia and hypertriglyceridemia. This paper summarizes existing knowledge on the role of ANGPTL3, 4, and 8 proteins in lipoprotein metabolism, the findings from clinical trials with evinacumab, a fully human ANGPTL3 mAb, and the place for this new agent in lipid-lowering therapy.

## 1. Homozygous Familial Hypercholesterolemia

Homozygous familial hypercholesterolemia (HoFH) affects an average of 1 in 300,000 subjects. It is a very rare genetic disorder of lipoprotein metabolism. It is caused by mutations in both alleles of the LDL receptor (LDLR) gene and less often by mutations in APOB, the ligand for LDLR and proprotein convertase subtilisin kexin type 9 (PCSK9), a protein that degrades LDLR [1]. Higher levels of low-density lipoprotein cholesterol (LDL-C) are characterized by genetic changes that show no expression of the LDL receptor (null homozygotes), compared to changes with two non-zero alleles or one zero and one non-null homozygotes, which only partially reduce the LDL receptor activity [2,3]. These mutations impair the function of the liver to remove LDL-C from the bloodstream, resulting in high total cholesterol and LDL-C [4]. In comparison, LDL particles that bind PCSK9 are targeted for lysosomal degradation and destruction. Loss-of-function mutations of the PCSK9 gene decrease the level of LDL-C and lower the risk of myocardial infarctions in white and black persons and reduce the risk of stroke in black persons. It can be concluded that PCSK9 inhibitors prevent an atherosclerotic cardiovascular event [1].

Patients with HoFH have very high levels of LDL-C from birth, which result in high risks of premature atherosclerosis and other cardiovascular diseases. Clinically, HoFH is characterized by an LDL-C level > 500 mg/dL (>13 mmol/L). Statins and lipid-lowering drugs are largely dependent on the activity of the LDL receptor; therefore, in patients with two null alleles, they may show diminished efficacy. Therefore, most patients with HoFH do not achieve guideline-recommended levels of LDL-C despite treatment with multiple agents [3]. Unfortunately, mutations in patients with familial hypercholesterolemia are associated with an increased probability (up to 3.8 times) of myocardial infarctions under the age of 55 years [5]. Early diagnosis of FH and follow-up, with comprehensive longitudinal care and particular emphasis on aortic valve obstruction and stenosis, are of key importance in the prevention of premature atherosclerotic cardiovascular disease (ASCVD) [2].

According to the 2019 guidelines of the European Society of Cardiology (ESC) and the European Society of Atherosclerosis (EAS), LDL-C levels should be below 1.42 mmol/L (55 mg/dL) in patients at very high risk of ASCVD, below 1.81 mmol/L (70 mg/dL) in patients at high risk, and below 2.59 mmol/L (100 mg/dL) in moderate-risk patients [5].

## 2. ANGPTL3, 4, and 8 Protein System-Characteristics and Role in Lipid Metabolism

The angiopoietin-like proteins (ANGPTLs) are a family of proteins consisting of members 1–8 of the angiopoietins, which differ in terms of tissue expression and regulation. They each consist of a common domain at the amino terminus (N-terminal), a coiled-coil domain (CCD), a fibrinogen-like domain (FLD) at the C-terminus of the carboxyl, and a linker region. Angiopoietin-8 differs from the other ANGPTLs in that it does not contain a fibrinogen-like domain at the C-terminus [6]. ANGPTL proteins belong to the vascular endothelial growth factor (VEGF) family and play various roles in biological and pathological processes, including hormone regulation, glucose metabolism, and insulin resistance [7].

ANGPTL3, ANGPTL4, and ANGPTL8 are most important in lipoprotein metabolism because they are responsible for the metabolism of triglycerides (TGs)—rich lipoproteins (chylomicrons, VLDL)—by inhibiting the activities of lipoprotein lipase (LPL), VLDL, and LDL mediated by the inhibition of endothelial lipase (EL) [6,8]. LPL activity is reduced by changing the conformation from homodimeric, which is biologically active, to biologically inactive, or monomeric. LPL is an enzyme produced in fat and muscle cells that limits the rate of hydrolysis of TG-rich lipoproteins to free fatty acids (FFA). When this process is disturbed, severe hypertriglyceridemia occurs in plasma [9]. The best-known ANGPTL is ANGPTL3, which was discovered in 1999. ANGPTL3 is produced in the liver. In the following year, 2000, ANGPTL4 was discovered, and it is produced in the liver, skeletal muscle, adipose tissue, gut, brain, and heart. Additionally, ANGPTL8 was discovered in 2012, and its main source is adipose tissue and the liver [10].

ANGPTL3, 4, and 8 control the availability of triglyceride-rich lipoproteins, LDL, and high-density lipoprotein cholesterol (HDL-C), depending on the nutritional status of the body, temperature, and physical activity, by regulating LPL secretion. LPL activity is increased after a meal, and triglycerides are stored in the white adipose tissue of WAT. In contrast, after a meal, LPL activity is reduced in the heart, brown adipose tissue, and skeletal muscle by ANGPTL3 and 8 (ANGPTL8 expression is especially increased). The opposite occurs during fasting, where LPL activity increases in the heart, brown adipose tissue, and skeletal muscle. In white adipose tissue, the activity of LPL during fasting is reduced by ANGPTL4 [11,12,13,14,15] (Figure 1).

ANGPTL3, apart from its effect on LPL, also reduces the activity of EL, which leads to a slowdown in the metabolism of triglyceride-rich lipoproteins [16].

## 3. ANGPTL3, 4, and 8 as Biomarkers of Cardiovascular Risk

Several publications have reported that ANGPTL3 deficiency protects against coronary artery disease (CAD). According to the research of Stitziel et al., in subjects with complete ANGPTL3 deficiency, the coronary arteries lacked atherosclerotic plaque [12]. Moreover, healthy patients showed lower concentrations of ANGPTL3 compared to patients who experienced myocardial infarctions (MIs) [12]. In patients with ANGPTL33 concentrations of 18–271 ng/mL, the risk of a heart attack was reduced by up to 29%. Researchers also demonstrated an association of the loss-of-function (LOF) mutation in *ANGPTL3* with the risk of CAD. The levels of low-density lipoprotein LDL-C, high-density HDL-C, and TGs are dependent on the LOF in *ANGPTL3*. Patients carrying the LOF mutation showed a 34% reduction in the risk of CAD compared to patients who did not carry the LOF mutation. In addition, patients with the LOF mutation showed 11% lower total cholesterol, 12% lower LDL, and 17% lower TG levels compared to those without the mutation. In addition to the fact that the loss of ANGPTL3 increases LPL activity, leading to a reduction in TGs and LDL-rich lipoproteins, it may affect insulin sensitivity and play an important role in glucose hemostasis [12].

In another study on the effects of ANGPTL3 and 4 on CAD, the team of Sun et al. presented the results of a study involving 305 patients. A high level of ANGPTL3 was closely related to the severity of atherosclerotic lesions in the coronary vessels, while the level of ANGPTL4 was reduced. The levels of these glycoproteins may have significant impacts on the development of CAD [17]. Another study showed the relationship between mutations inactivating the ANGPTL4 gene on the risk of ischemic heart disease. This study included over 42,000 subjects. Dewey et al. in 2017 proved that the reductions in the levels of TGs, total cholesterol, and LDL-C were caused by the inactivation of ANGPTL4 through the heterozygous mutation E40k. Patients with this ANGPTL4 mutation showed a 19% lower risk of coronary heart disease [18]. In a similar study led by Stitziel et al., patients with the E40K ANGPTL4 mutation showed about 35% lower TG concentration. Additionally, the risk of coronary heart disease was 53% lower. However, no significant effect of ANGPTL4 p.E40K on LDL-C was observed [19]. The research team of Gusarova et al. showed the effect of the E40K ANGPTL4 mutation on the reduction of the risk of type-2 diabetes by 12%. This study was conducted on 58,000 participants in the DiscovEHR Study [20]. Similar results were presented by the team of Klarin et al. in a study of 310,000 subjects. The effect of the loss-of-function (LOF) ANGPTL4 mutation on the risk of ischemic heart disease and type-2 diabetes was assessed. It was shown that the risk of ischemic heart disease was reduced by 16% and the risk of type 2 diabetes was reduced by 12% [21].

## 4. Evinacumab-Structure and the Mechanism of Action

Evinacumab (Evkeeza^®^; formerly RENG1500) is a fully human monoclonal antibody, inhibiting circulating ANGPTL3, which was invented by Regeneron Pharmaceuticals Inc. [3] and manufactured with the use of the cell culture method with genetically engineered recombinant Chinese hamster ovary cells [22]. Evinacumab is an IgG4 monoclonal antibody consisting of two disulfide-linked human heavy chains (453 amino acids each) and human kappa light chains (214 amino acids). Heavy chains are covalently linked by disulfide bonds to light chains [22].

Evinacumab was approved by the US Food and Drug Administration on February 2021 and the European Medicines Agency (EMA) in June 2021 and is now available on the market under the trade name Evkeeza to treat adult and adolescent patients (≥12 years) with homozygous familial hypercholesterolemia [23]. The recommended dose of this new drug is 15 mg/kg, administered by intravenous infusion (IV) over one hour once monthly [23].

After administration, evinacumab binds its target, ANGPTL3, and inhibits its function, leading to increased LPL and EL activities and lower TG, LDL-C, and HDL-C plasma levels [24]. The mechanism associated with the reduction of LDL-C by evinacumab is not fully known; however, this effect is independent of the LDL receptor and, thus, probably due to the promotion of very-low-density lipoprotein (VLDL) processing and the upstream clearance of LDL formation [12,13,14,15,24]. The mechanism of the action of evinacumab is presented in Figure 2.

### Clinical Trials and Scientific Research

In the first phase, phase 1, a randomized, placebo-controlled, double-blind clinical trial with evinacumab (NCT01749878) was performed with subjects with hypertriglyceridemia (HTG) to evaluate its safety, tolerability, and bioeffect. A total of 83 healthy volunteers with fasting triglyceride levels of 150–450 mg/dL (1.7 ≤ 5.1 mmol/L) or LDL-C levels of ≥100 mg/dL (2.6 mmol/L) were enrolled in cohort A, and each received a single dose of evinacumab administrated subcutaneously (SC) (75, 150, and 250 mg) or intravenously (IV) (5 mg/kg, 10 mg/kg, and 20 mg/kg) versus placebo [18]. Participants (*n* = 7) allocated to cohort B (moderate HTG) had TG levels of >150 and ≤450 mg/dL, and each received evinacumab IV at a dose of 10 mg/kg or placebo [19]. Cohort C (severe HTG) participants (*n* = 9) had LPL pathway sequence variations and TG levels of >1000 mg/dL and received evinacumab at a dose of 250 mg SC or 20 mg/kg IV versus placebo [18]. Evinacumab caused a dose-dependent reduction in lipids levels. The greatest reductions of TG, LDL-C, and HDL-C were 76.0% (day 4) (95% CI: −97.29, −62.02; *p* < 0.0001); 23.2% (day 15) (95% CI: −7.59, −38.80; *p* = 0.0047); and 18.4% (day 15) (95% CI: −5.96, −30.77; *p =* 0.0049), respectively, and these reductions were noted in cohort A participants who received a dose of 20 mg/kg IV [18].

There were no treatment discontinuations due to adverse safety events in all cohorts (A–C) [18,25]. Treatment-emergent adverse events occurred in 51.6% of cohort A (vs. 42.9% placebo) and 100% of cohorts B–C (vs. 81.8% placebo) [18,25]. The most frequent adverse events (AEs) observed in evinacumab-treated patients (cohort A) were headache (11% vs. 0% placebo), upper respiratory tract infection (6.5% vs. 4.0% placebo), increased alanine aminotransferase (11.3% vs. 0% placebo), and increased aspartate aminotransferase (4.8% vs. 0% placebo).

In cohort B, evinacumab caused a maximal reduction of TGs by 81.8% (vs. 20.6% placebo) and VLDL-C by 82.2% (vs. 0.8% placebo) on day 4. Treatment with evinacumab-induced wide-ranging responses in subjects with severe hypertriglyceridemia (cohort C) showed a TG reduction of 0.9 to 93.2% on day 3 [25].

In cohort C, evinacumab IV at a dose of 20 mg/kg caused a maximum mean reduction in VLDL-C of 64.9% (vs. 42.0% placebo) on day 22, while evinacumab SC at a dose of 250 mg resulted in a maximum reduction of 37.8% (vs. increase of 18.4% placebo) on day 8. The levels of LDL-C after treatment with evinacumab increased in participants with moderate and severe HTG (cohorts B and C), which was explained by the authors as a reason for the enhanced conversion of VLDL and IDL to LDL [25].

Another phase 1 clinical trial (NCT02107872) with subjects with triglycerides >150 and ≤500 mg/dL and LDL-C ≥ 100 mg/dL was conducted to assess the effect of multi-doses of evinacumab [26]. A total of 56 participants were enrolled, and each was assigned to one of the six cohorts as follows: evinacumab SC at doses of 150, 300, or 450 mg QW, 300 or 450 mg Q2W, or IV doses of 20 mg/kg Q4W up to day 56 versus placebo with 183 days of follow-up [26]. Treatment-emergent adverse events occurred in 67.7% of patients with evinacumab SC (vs. 75% placebo) and 85.7% of the subjects with evinacumab IV (vs. 50% placebo). No serious treatment-emergent adverse events, events related to death, or treatment discontinuation were reported. The most commonly reported adverse events were headache (42.9% vs. 0% placebo) in the group with evinacumab IV and nausea (13% vs. 0% placebo) in the group with evinacumab SC [26].

Dose-dependent reductions in triglycerides were observed, with a maximum reduction at a dose of 20 mg/kg Q4W IV by 88.2% at day 2 (*p* = 0.0003). Other lipids such as non-HDL, apoB, LDL-C, and total cholesterol were maximally reduced at dose 20 mg/kg Q4W IV by 45.8% (day 36) (*p* < 0.0001), 30.7% (day 57) (*p* < 0.0001), 25.1% (day 57) (*p* = 0.0074), and 33.8% (day 57) (*p* < 0.0001), respectively [26].

The pharmacokinetics, tolerability, safety, and lipid-lowering effect of evinacumab were evaluated in phase 1 with a randomized, double-blind, placebo-controlled clinical trial (NCT03146416) with healthy Caucasian and Japanese volunteers with LDL-C concentrations between 100 and 160 mg/dL (2.6–4.1 mmol/L). A total of 96 enrolled participants was divided into four cohorts: a single dose of 300 mg of evinacumab SC; 300 mg (SC) once weekly for eight doses; 5 mg/kg IV once every 4 weeks for two doses; and 15 mg/kg IV once every 4 weeks for two doses. Each cohort comprised 12 Japanese and 12 Caucasians, who each received an investigational drug or placebo with a 24-week follow-up [3]. The results of the study indicated that the safety of evinacumab treatment at all doses and routes of administration in both ethnic groups was comparable with that of the placebo. Observed adverse events related to the treatment were nausea, fatigue, nasopharyngitis, upper respiratory infection, rhinitis, back pain, headache, and tension headache. In cohorts with IV evinacumab administration, adverse events occurred in 15 subjects (41.7%) with the drug versus 6 subjects (50%) in the placebo group. In cohorts with evinacumab SC, adverse events were noted in 19 subjects (52.8%) with the drug versus 3 subjects (25.0%) with a placebo. No severe or serious adverse event was observed. Moreover, the pharmacokinetic and pharmacodynamic profiles of evinacumab were similar in both ethnicities in all treatment groups [4]. Evinacumab caused dose-dependent reductions in LDL-C, TG, and apoB levels. The administration of evinacumab IV at a dose of 15 mg/kg every 4 weeks in two doses caused the greatest lipid-lowering power. In addition, the reductions of LDL-C, triglycerides, non-HDL cholesterol, HDL cholesterol, total cholesterol, apoB, apoA-I, apoC-III, and Lp(a) in plasma after 8 weeks were 40.2%, 63.1%, 44.2%, 23.8%, 40.2%, 37.4%, 33.5%, 77.1%, and 22.2%, respectively [4].

The safety and efficacy of evinacumab in patients with refractory hypercholesterolemia were assessed in the phase 2 clinical trial (NCT03175367) [5]. A total of 272 subjects with HeFH or without HeFH with refractory hypercholesterolemia and who had LDL-C ≥ 70 mg/dL with ASCVD or LDL-C ≥ 100 mg/dL without ASCVD were enrolled. Participants received evinacumab SC in different doses of 450 mg QW, 300 mg QW, or 300 mg Q2W vs. placebo or evinacumab IV in a dose of 15 mg/kg Q4W or 5 mg/kg Q4W versus placebo [5].

The most common side effects reported by the participants during the trial in the group treated with SC evinacumab were urinary tract infections (11% vs. 8%), injection-site erythema (6% vs. 3%), myalgia (5% vs. 0%), and arthralgia (5% vs. 3%), whereas the group receiving IV evinacumab had nasopharyngitis (12% vs. 6%), dizziness (7% vs. 0%), nausea (7% vs. 0%), abdominal pain (6% vs. 0%), back pain (7% vs. 6%), fatigue (7% vs. 6%), and arm or leg pain (7% vs. 6%). Serious adverse events were noted in 5–8% (8% placebo) in the group with SC evinacumab and in 6–16% (versus 3% placebo) in the group with IV evinacumab [5]. Treatment with evinacumab caused a reduction in the level of LDL-C. In the group with SC administration, the reductions were between 38.5% (95% CI: –56.5, –20.6; *p* < 0.001) and 56.0% (95% CI: –73.7, –38.3; *p* < 0.001), while the group with IV administration had reductions from 24.2% (95% CI: –42.6, –5.7) to 50.5% (95% CI: –68.4, –32.6; *p* < 0.001) depending on the dose used. The greatest decrease in LDL-C was observed with a dose of 450 mg QW of evinacumab SC (56.0%) (95% CI: −73.7, −38.3; *p* < 0.001) [5].

Several clinical trials evaluating the efficacy and safety of evinacumab in patients with homozygous familial hypercholesterolemia have been completed. A phase 2, open-label, proof-of-concept clinical trial (NCT02265952) included nine HoFH participants receiving maximum-tolerated lipid-lowering treatment. All subjects were administered evinacumab at doses of 250 mg SC and 15 mg/kg IV after 2 weeks [27]. All participants reported at least one adverse event, but no event led to treatment discontinuation. Injection-site reactions, myalgia, hot flush, and epistaxis were reported as treatment-emergent AEs, and each of these AE was observed by only one patient [27]. Evinacumab caused reductions in LDL-C, non-HDL cholesterol, triglycerides, apolipoprotein B, and HDL cholesterol by mean values of 49 ± 23%, 49 ± 22%, 47 ± 17%, 46 ± 18%, and 36 ± 16%, respectively [27].

Banerjee et al. analyzed the effects of evinacumab on LDLR activity in lymphocytes drawn from participants in the above-mentioned study (NCT02265952). Obtained results suggested that the inhibition of ANGPTL3 by evinacumab in humans lowers LDL-C through a mechanism independent of the LDLR [28].

A small kinetic study (NCT04722068) with four subjects already participating in the study (NCT02265952) was conducted to investigate the apolipoprotein B (apoB) containing lipoprotein kinetic parameters before and after treatment with evinacumab [29]. A stable isotope of Leucine was measured in VLDL (very-low-density lipoprotein), IDL (intermediate-density lipoprotein; VLDL remnants), and LDL before and after IV administration of evinacumab at a dose of 15 mg/kg. Evinacumab decreased LDL-C by 59 ± 2% and increased IDL-apoB and LDL-apoB fractional catabolic rates in all four HoFH participants by 616 ± 504% and 113 ± 14%, respectively. The VLDL-apoB production rate was reduced in two of the four subjects. These results suggest that the mechanism of lowering LDL-C by evinacumab is associated with the increased clearance of apoB-containing lipoprotein [29].

A phase 3, double-blind, placebo-controlled clinical trial (ELIPSE HoFH, NCT03399786) enrolled 65 subjects with HoFH on stable lipid-lowering therapy. Ninety-three percent of the subjects were on statin therapy, and the majority of the patients were receiving high-intensity statin (77%). Moreover, trial patients’ background therapy also included PCSK9 inhibitor (77%), ezetimibe (75%), lomitapide (25%), and apheresis (34%). Of the patients, 63% used at least three different lipid-lowering drugs. Participants received evinacumab IV 15 mg/kg every 4 weeks for 24 weeks or a placebo. Treatment with evinacumab caused significant decreases in plasma levels of LDL-C, total cholesterol, non-HDL, HDL, triglycerides, apoB, apoC-III, and Lp(a) by 47.1% (95% CI: −65.0, −33.1; *p* < 0.001), 47.4% (95% CI: −58.7 to −38.1; *p* < 0.001), 49.7% (95% CI: −64.8 to −38.5; *p* < 0.001), 29.6%, 55.0% (95% CI: −65.6, −35.2), 41.4% (95% CI: −48.6, −25.2; *p* < 0.001), 84.1% (95% CI: −103.5 to −76.5), and 5.5% (95% CI: −15.7, 12.0), respectively [3]. A 50% reduction in plasma LDL-C concentration was achieved in 56% (*p* = 0.003) of patients; moreover, 28% of patients had plasma LDL-C levels below 70 mg/dL. The efficacy of the LDL-C reduction with evinacumab was independent of the type of LDL receptor gene mutations (non/null or null/null). Adverse events were comparable in the group with evinacumab and the placebo group. An influenza-like illness occurred in 5 of 44 patients (11%) vs. 0 in the placebo group. No events related to death or treatment discontinuation were reported [3].

Additionally, the study of Reeskamp et al. analyzed whether intensive lipid-lowering therapy with evinacumab might result in plaque regression. Using computed tomography (CT) coronary angiography, soft plaque regression occurred in the coronary arteries of two participants, ages 12 and 16, of a clinical trial (NCT03399786) with HoFH and null/null LDLR variants. Total plaque volumes were reduced by 76% and 85%, respectively, after 6 months of evinacumab treatment [30].

Most of the completed clinical trials with evinacumab had small sample sizes. Jin et al. conducted a meta-analysis of five randomized controlled trials with 568 participants and revealed that treatment with evinacumab was safe and caused a reduction of LDL-C, TG, and HDL cholesterol by 33.12% (95%: 248.639%, 217.606%, *p* < 0.0001), 50.96% (95% CI: 256.555%, 245.362%; *p* < 0.0001), and 12.77% (95% CI: 216.359, 29.186%, *p* < 0.0001), respectively [31]. The results of this meta-analysis found evinacumab as a possible valuable therapeutic in the management of hypercholesterolemia.

The efficacy of evinacumab is under clinical trial phase 2 (NCT04863014) [32] and two phase 3 studies (NCT03409744; NCT04233918) [33,34] (Table 1).

A phase 2, double-blind clinical trial NCT04863014 started on July 2021, and the estimated study completion date is February 2023. The study enrolled 21 adult participants with severe hypertriglyceridemia and will assess the efficacy and safety of evinacumab for the prevention of recurrent acute pancreatitis [32].

A phase 3, open-label clinical trial (NCT04233918) with 20 pediatric participants (5–11 years) with HoFH aims to evaluate the efficacy and safety of evinacumab. The estimated completion date is May 2023 [34]. A phase 3, open-label clinical trial (NCT03409744) enrolled 116 adolescent participants (≥12 years) with HoFH. The aim of the study is to evaluate the long-term safety and efficacy of evinacumab. Initial results are expected in January 2023 [30]. Completed and ongoing clinical trials with evinacumab are summarized in Table 1.

## 5. Perspectives of Evinacumab in Clinical Lipidology

Guidelines of the Polish Lipid Association (PoLA) from 2021 regarding lipid-lowering treatment place evinacumab in the group of drugs supporting the therapy of familial hypercholesterolaemia [35]. In the guidelines of the European Society of Cardiology (ESC) and the European Atherosclerotic Society (EAS) from 2019 regarding the management of dyslipidemias, evinacumab is characterized in the context of new approaches to reduce triglyceride-rich lipoproteins and their remnants [36]. There are currently no official recommendations regarding evinacumab and its place in the treatment of lipid disorders.

Evinacumab may be of great help in lowering LDL-C levels in patients with HoFH in whom the available treatment (statins, ezetimibe, PCSK9 inhibitors) did not achieve the therapeutic goal [37]. Experts have also indicated that evinacumab may be an important drug in the context of the reduction of triglycerides associated with ASCVD residual risk, especially in patients with diabetes [38].

The future of evinacumab seems to be slightly different from that of other drugs intended for severe hypercholesterolemia, such as PCSK9 inhibitors (alirocumab, evolocumab) and inclisiran. These examples do not relate to hypertriglyceridemia. However, new clinical trials are needed to expand our experience of combining evinacumab with ezetimibe, bempedoic acid, or even PCSK9 modulators in patients currently treated with LDL apheresis. The authors of this article pose some open questions about the future of evinacumab: (1) Will evinacumab be a better choice in the future than the concept of PCSK9 blockade? (2) Will this change the way we currently treat hypertriglyceridemia? (3) What will be the optimal algorithm for introducing this drug to the family of old and new hypolipemic therapies?

The approval and availability of evinacumab in the lipid-lowering armamentarium will undoubtedly constitute significant progress, providing a drug with high efficacy in not only hypercholesterolaemia but also hypertriglyceridemia. The authors believe that, in addition to the expected breakthrough therapies in the field of lowering lipoprotein (a) concentrations, i.e., the intensively studied pelacarsen and olpasiran or SLN360, the entry into the pharmaceutical armamentarium of evinacumab may become the most important event of lipoprotein pharmacotherapy in the current decade. An antisense oligonucleotide (ASO) that reduces ANGPTL3 synthesis in the liver-vupanorsen is also in clinical trials [39]. During the Congress of the European Atherosclerosis Society (EAS) in 2022, a liver-specific treatment of the target protein ANGPTL4 was also presented [40]. Thus, the system of proteins ANGPTL3, ANGPTL4, and ANGPTL8 is currently a widely studied point of the pharmacological action of lipid-lowering drugs.

## 6. Conclusions

The above-mentioned findings from clinical trials (Table 1) and scientific research proved that evinacumab is effective and safe as an add-on treatment in the management of HoFH. Evinacumab decreases the level of LDL cholesterol by approximately 50% in individuals with maximally tolerated lipid-lowering therapy, and its mechanism is independent of residual LDLR activity [3,27]. Moreover, it was indicated that evinacumab might also cause plaque regression; [30] however; this finding should be validated in randomized, placebo-controlled trials in large groups of patients.

Traditional lipid-lowering therapies, such as statins and PCSK9 inhibitors [8,41,42,43], which upregulate the LDLR pathway, are ineffective or less effective in individuals with two null alleles present in HoFH [44]. In contrast, evinacumab lowers LDL-C levels independent of LDLR activity, so it can be considered a major tool in the armamentarium of patients with HoFH who failed to achieve their minimal guideline-recommended LDL-C goals, despite receiving multiple classes of lipid-lowering therapies and LDL apheresis, or as an alternative for patients who do not tolerate or have no access to apheresis or lomitapide [45,46].

Unfortunately, access to evinacumab, similar to other newly approved potent lipid-lowering therapeutics (PCSK9 inhibitors, lomitapide), is commonly restricted by regulatory approval and high cost [47,48].

Although evinacumab is approved for use by the FDA and EMA, the long-term effect of its action still needs to be studied. Currently, the long-term effect of evinacumab is being studied in an ongoing clinical trial, and the first results are expected in 2023.

## Figures and Tables

**Figure 1 jcm-12-00168-f001:**
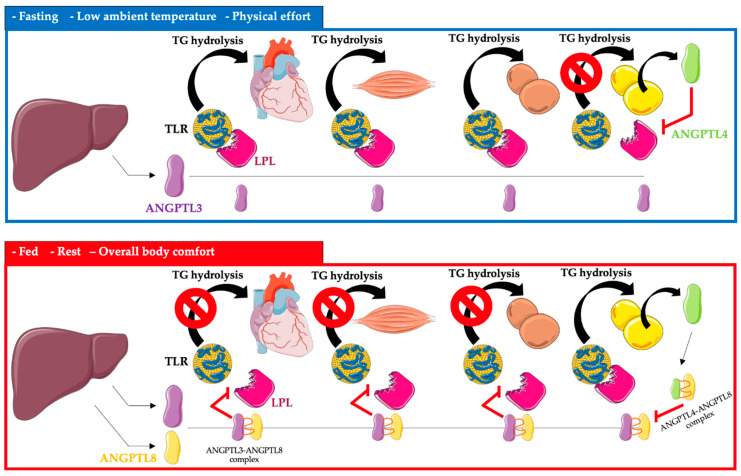
Regulation of triglyceride metabolism in heart, muscle, brown adipose tissue, and white adipose tissue by ANGPTL3, ANGPTL4, and ANGPTL8. Abbreviations: TG—triglyceride; TLR—triglyceride-rich lipoprotein; LPL—lipoprotein lipase; ANGPTL3—angiopoietin-like protein 3; ANGPTL4—angiopoietin-like protein 4; ANGPTL8—angiopoietin-like protein 8. The following was used in the preparation of the figure: https://smart.servier.com (free-access; 20 October 2022).

**Figure 2 jcm-12-00168-f002:**
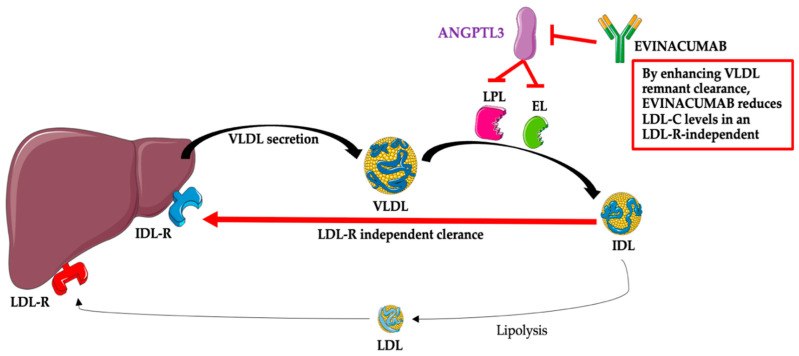
Evinacumab—mechanism of action. Abbreviations: ANGPTL3—angiopoietin-like protein 3; EL—endothelial lipase; IDL—intermediate-density lipoprotein (VLDL remnants); IDL-R—intermediate-density lipoprotein receptor (VLDL remnant receptor); LDL—low-density lipoprotein; LDL-C—low-density lipoprotein; LDL-R—low-density lipoprotein receptor; LPL—lipoprotein lipase; VLDL—very-low-density lipoprotein. The following was used in the preparation of the figure: https://smart.servier.com (free-access; 20 October 2022).

**Table 1 jcm-12-00168-t001:** Summary of the results of clinical trials with evinacumab.

Clinical Phase/Status	NCT Identification Number	Population/*N*	Duration	Dose/Treatment Arms	Key Results	Safety	Ref.
Phase 1 Completed	NCT01749878	HTG Cohort A: 150 < TG ≤ 450 mg/dL or LDL-C ≥ 100 mg/dL *N* = 83 Cohort B: 450 ≤ TG < 1500 mg/dL *N* = 7 Cohort C: LPL pathway sequence variations, TG > 1000 mg/dL *N* = 9	126 days	Single dose: Cohort A: 75, 150, 250 mg SC or 5, 10, 20 mg/kg IV vs. placebo Cohort B: 10 mg/kg IV vs. placebo Cohort C: 250 mg SC or 20 mg/kg IV vs. placebo	Max reduction: Cohort A: At a dose of 20 mg/kg IV TG: 76.0% LDL-C: 23.2% HDL: 18.4% Cohort B TG: 81.8% VLDL-C: 82.2% Cohort C TG: 0.9 to 93.2%, VLDL-C: 64.9% (IV 20 mg/kg), 37.8% (SC 250 mg)	TEAEs Cohort A 51.6% vs. 42.9% placebo Cohorts B and C 100% vs. 81.8% placebo Frequent AE (cohort A): headache (11% vs. 0% placebo), upper respiratory tract infection (6.5% vs. 4.0% placebo), increased alanine aminotransferase (11.3% vs. 0% placebo), increased aspartate aminotransferase (4.8% vs. 0% placebo)	[18,25,26]
Phase 1 Completed	NCT02107872	HTG: 150< TG ≤500 mg/dL and LDL-C ≥ 100 mg/dL *N* = 56	183 days	Multiple doses: **SC**: 150, 300, 450 mg QW or 300, 450 mg Q2W **IV**: 20 mg/kg Q4W up to day 56 vs. placebo	Max reduction at IV 20 mg/kg Q4W TG: 88.2% (day 2) non-HDL: 45.8% (day 36) apoB: 30.7% (day 57) LDL-C: 25.1% (day 57) total cholesterol: 33.8% (day 57)	TEAEs **SC**: 51.6% vs. 42.9% placebo **IV**: 85.7% vs. 50% placebo Common AE: **SC**: nausea (13% vs. 0% placebo) **IV**: headache (42.9% vs. 0% placebo); No SAE	[26]
Phase 2 Completed	NCT02265952	HoFH *N* = 9	26 weeks	Two doses: 250 mg SC on day 1 and 15 mg/kg IV on week 2	Max reduction (week 4): LDL-C: 49 ± 23% non-HDL-C: 49 ± 22% TG: 47 ± 17% apo B: 46 ± 18% HDL-C: 36 ± 16%	TEAEs: Injection-site reactions (11%), myalgia (11%), hot flush (11%), epistaxis (11%); No SAE	[27,28]
Phase 1 Completed	NCT03146416	Healthy Japanese and Caucasian *N* = 96	24 weeks	Cohorts: I: single dose of 300 mg (SC) II: 300 mg (SC) QW for eight doses III: 5 mg/kg (IV) IV: 15 mg/kg (IV) Q4W for two doses vs. placebo	Max reduction, 15 mg/kg (IV) Q4W for two doses (week 8): LDL: 40.2% TG: 63.1% non-HDL: 44.2% HDL: 23.8% total cholesterol: 40.2% apoB: 37.4% apoA-I: 33.5% apoC-III: 77.1% Lp(a): 22.2%	TEAEs SC: 52.8% vs. 25.0% placebo IV: 41.7% vs. 50% placebo Common TEAEs: nausea, fatigue, nasopharyngitis, upper respiratory infection, rhinitis, back pain, headache, tension headache; No SAE	[4]
Phase 2 Completed	NCT03175367	Hypercholesterolemia: HeFH or non-HeFH with ASCVD ASCVD: LDL-C ≥ 70 mg/dL or non-ASCVD: LDL-C ≥ 100 mg/dL *N* = 272	16 weeks	SC: 450 mg QW, 300 mg QW, or 300 mg Q2W vs. placebo; IV: 15 mg/kg Q4W or 5 mg/kg Q4W vs. placebo	LDL-C reduction: SC: 38.5% to 56.0%, IV: 24.2% to 50.5%	SC AE: 68–82% vs. 54% placebo; urinary tract infection (11% vs. 8%), injection-site erythema (6% vs. 3%), arthralgia (5% vs. 3%), myalgia (5% vs. 0%) SAE: 5–8% vs. 8% IV AE: 75–84% vs. 70% placebo; abdominal pain (6% vs. 0%), back pain (7% vs. 6%), dizziness (7% vs. 0%), fatigue (7% vs. 6%), pain in an arm or leg (7% vs. 6%), nausea (7% vs. 0%), and nasopharyngitis (12% vs. 6%); SAE: 6–16% vs. 3%	[5]
Phase 2 Completed	NCT03452228	sHTG, TG values ≥ 500 mg/dL (5.6 mmol/L) *N* = 52	24 weeks	Evinacumab IV vs. placebo	NA	NA	
Phase 3 Completed	NCT03399786	HoFH LDL-C ≥ 70 mg/dL (1.8 mmol/L) *N* = 65	24 weeks	Evinacumab IV 15 mg/kg Q4W vs. placebo	Reduction: LDL-C: 47.1%, total cholesterol: 47.4%, non-HDL: 49.7%, HDL: 29.6%, TG: 55.0%, apoB: 41.4%, apoC-III: 84.1% Lp(a): 5.5%	AE: 66% vs. 81% placebo, SAE: 5% vs. 0 placebo	[3]
Kinetics test	NCT04722068	HoFH *N* = 4	8 weeks	Evinacumab IV 15 mg/kg	Decrease: LDL-C: 59 ± 2%, Increase: IDL-ApoB: 616 ± 504%, LDL-ApoB: 113 ± 14%,	NA	[29]
Phase 3 Ongoing	NCT03409744	HoFH, adolescent subjects (≥12 years) *N* = 116	192 weeks	Evinacumab IV	Study completion date: January 2023		[23]
Phase 3 Ongoing	NCT04233918	HoFH, pediatric subjects (5–11 years), LDL-C > 130 mg/dL *N* = 20	24 weeks	Part A: single IV dose; Part B: IV dose Q4W until week 20; Part C: IV dose Q4W	Study completion date: May 2023		[23]
Phase 2 Ongoing	NCT04863014	sHTG, adult subjects (18–80 years) TG > 880 mg/dL (10 mmol/L) or >500 mg/dL (5.6 mmol/L); *N* = 21	52 weeks	IV Q4W vs. placebo	Study completion date: February 2023		[23]

ASCVD, atherosclerotic cardiovascular; AE, adverse event; CVD, cardiovascular disease; HoFH, homozygous familial hypercholesterolemia; HTG, Hypertriglyceridemia; IV, intravenous; LPL, lipoprotein lipase; NA, no data are available; SAE, serious adverse event; SC, subcutaneous; sHTG, severe hypertriglyceridemia disease; QXW, once every X weeks; TEAEs, treatment-emergent adverse events.

## Data Availability

Not applicable.

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
