# Peer review of "Evinacumab, an ANGPTL3 Inhibitor, in the Treatment of Dyslipidemia"

_jcm, 2022, doi:10.3390/jcm12010168_

Round 1
Reviewer 1 Report
The paper presented is well written and the topic is very interesting.
Here few minor comments:
- Table 1 summarizing all the studies published needs to be improved.
- Line 43-44: "Low LDL-C levels and the risk of myocardial infarction in whites and blacks and stroke in blacks are associated with loss-of-function mutations in PCSK9." Can you reframe this sentence to make it more understandable?
- Line 295 "The most common side effects reported by the participants during the trials were non- specific" what the authors mean with side effects were non-specific? can you reframe the sentence?
Author Response
I am very grateful to the Reviewers for their thorough reading of my paper and received remarks and suggestions. We made a meticulous revision of the manuscript, following all the comments made by the Reviewers. Please find below our responses and details of the changes we made in the text.
Reviewer #1:
The paper presented is well written and the topic is very interesting.
Here few minor comments:
- Table 1 summarizing all the studies published needs to be improved.
Response:
The table was improved to be clearer for the reader. Abbreviations below the table have been arranged in alphabetical order.
- Line 43-44: "Low LDL-C levels and the risk of myocardial infarction in whites and blacks and stroke in blacks are associated with loss-of-function mutations in PCSK9." Can you reframe this sentence to make it more understandable?
Response:
Sentence was reframed:
Loss-of-function mutations of the PCSK9 gene decrease the level of LDL-C and low the risk of myocardial infarction in whites and blacks and reduce the risk of stroke in blacks.
- Line 295 "The most common side effects reported by the participants during the trials were non- specific" what the authors mean with side effects were non-specific? can you reframe the sentence?
Response:
The sentence was reframed:
The most common side effects reported by the participants during the trial in the group treated with SC evinacumab were: urinary tract infection (11% vs 8%), injection-site erythema (6% vs 3%), myalgia (5% vs. 0%), and arthralgia (5% vs 3%), whereas in the group receiving IV evinacumab were: nasopharyngitis (12% vs 6%), dizziness (7% vs 0%), nausea (7% vs. 0%), abdominal pain (6% vs. 0%), back pain (7% vs 6%), fatigue (7% vs 6%), and pain in an arm or leg (7% vs 6%).
Reviewer 2 Report
Congratulations to the authors for this work.
Remarks:
- in the present paper, it is not mentioned whether the study is registered in the Clinical Studies Database and the endorsement of the ethics committee is not mentioned.
- the lack of a statistical interpretation of the obtained data is felt, so I would suggest the authors to add a statistical interpretation to the results and discussions.
Author Response
I am very grateful to the Reviewers for their thorough reading of my paper and received remarks and suggestions. We made a meticulous revision of the manuscript, following all the comments made by the Reviewers. Please find below our responses and details of the changes we made in the text.
Reviewer #2:
Congratulations to the authors for this work.
Remarks:
- in the present paper, it is not mentioned whether the study is registered in the Clinical Studies Database and the endorsement of the ethics committee is not mentioned.
Response:
It is a review study, therefore no registration or ethics committee information/approval is required.
- the lack of a statistical interpretation of the obtained data is felt, so I would suggest the authors to add a statistical interpretation to the results and discussions.
Response:
Data of the available statistical analysis was added to the results of the studies. Studies with a small sample sizes have no performed formal statistical testing and the results were summarized as mean±SD.
Reviewer 3 Report
In this manuscript, Sosnowska et.al., reviewed evinacumab,(Evkeeza) in Familial Hypercholestrelemia. Manuscript was well-written. I do not have a recommendation for the article at the moment.
Author Response
Dear Reviewer,
Thank you very much for such a positive opinion about our work.
Best wishes,
Authors